# Experiments and Numerical Simulations of the Annealing Temperature Influence on the Residual Stresses Level in S700MC Steel Welded Elements

**DOI:** 10.3390/ma13225289

**Published:** 2020-11-22

**Authors:** Tomasz Kik, Jaromír Moravec, Martin Švec

**Affiliations:** 1Department of Welding Engineering, Silesian University of Technology, Konarskiego 18A, 44-100 Gliwice, Poland; 2Department of Engineering Technology, Technical University of Liberec, 461 17 Liberec, Czech Republic; jaromir.moravec@tul.cz (J.M.); martin.svec@tul.cz (M.Š.)

**Keywords:** S700MC, welding thermal cycle, annealing, FEM, numerical analyses, stresses distribution

## Abstract

The article presents the results of research on the influence of temperature and time changes of the annealing process on the values and distribution of stresses in the simulated heat-affected zone of S700MC steel welded joints. For this purpose, tests were carried out on a thermal cycle simulator, as well as heating the prepared samples in accordance with the recorded welding thermal cycles, and then annealing at temperatures from 200 to 550 °C. The stresses values in the tested samples before and after the annealing process were measured by using X-ray diffraction (XRD). The performed tests were verified with the results of numerical analyses using the finite element method (FEM) performed in the VisualWeld (SYSWELD) environment as, on the one hand, the verification of the obtained results, and, on the other hand, the source of data for the development of a methodology for conducting analyses of heat treatment processes of S700MC steel welded structures. Also presented are three examples of numerical analyses for Gas Metal Arc (GMAW), laser and hybrid welding and then the annealing process of the obtained joints at selected temperatures. The main purpose of the work was to broaden the knowledge on the influence of annealing parameters on the values and distribution of stresses in welded joints, but also to signal the possibility of using modern software in engineering practice.

## 1. Introduction

Today it is impossible to imagine a world without welded structures. From household appliances to huge vessels or oil rigs, construction elements connected using many known welding methods are used everywhere. When designing these modern constructions, we strive for ever higher efficiency, demanding strength, lower weight and high durability from the materials used during operation, sometimes in very diverse environmental conditions [1,2,3]. Due to the intensive development of the iron metallurgy and the use of modern, highly specialized production processes, it is now possible to produce steel materials with properties compliant with the requirements of the engineering materials market, and not yet available for this group of materials. A complicated process of thermomechanical rolling based on applying the deformation of the material in a certain temperature ranges, leading to the achievement of a state characterized by specific, higher properties that cannot be achieved by steel forming process alone [4,5]. Thanks to this, it is possible to reduce the weight while maintaining the strength of the materials, which brings enormous economic benefits, resulting not only from the costs of the material itself, but also its preparation for welding, processing and also transport. The properties of thermomechanically rolled steels (TMCP—Thermo Mechanical Controlled Processing) have been enthusiastically received by industries such as automotive, machinery, construction and shipbuilding. They are used for statically and dynamically loaded components, especially in the automotive industry in the production of axles and chassis of cars and trucks, but also for components of cranes and other structural units [2,6,7,8].

The advantages of these group of steels are associated with much higher production requirements. In addition to the aforementioned thermomechanical treatment, the content of micro-alloying additives such as niobium, vanadium and titanium in a strictly defined content, in combination with thermomechanical treatment, enables grain refinement and precipitation hardening to take place [9,10,11,12]. It is also to be expected that the complicated manufacturing process of these steels will be a problem if they are subjected to the welding thermal cycle. A different method of production compared to conventional techniques, as is easy to guess, has a significant influence on the properties of TMCP steel welded joints [9,12]. The research results available in the literature indicate as the main problems that arise during the welding process:-loss of the initial properties during and after the welding process, which may cause partial dissolution of fine-dispersion strengthening precipitates (carbides, Nb, Ti, V carbonitrides) and their uncontrolled re-precipitation-excessive growth of strengthening precipitates and loss of their ability to inhibit grain growth,-transfer of microalloying elements to the weld and the difference in the carbon equivalent between the base material and the weld [5,13,14,15,16].

However, this is not the only problem faced by welding engineers using these TCMP steels. Welded structures, usually consisting of dozens of welds, very often require heat treatment to reduce the level of stresses remaining after the welding process and to homogenize their distribution. This is of great importance if we consider the fact that the unfavorable changes caused by welding thermal cycles may increase the tendency to brittle and fatigue cracking [16,17]. Also, in this case, TMCP steels require the use of very precise settings of the process itself due to the risk of loss of strength and plastic properties [10,14,15,18]. From the product data sheets, it can be determined that normalizing and stress relieving annealing at temperatures above 580 °C with long holding times results in a reduction of the yield point and tensile strength compared to the “as delivered” condition [19]. The available research results in this field also indicate that the annealing process conducted in a wide range of temperatures (from 100 to 1300 °C) showed that the stability of the bainitic structure is maintained up to the temperature of 600 °C. Above this temperature, a slight grain growth is noted and exceeding the processing temperature of 1000 °C causes an increase in the amount of ferrite in the grain structure and size. The steel retains its strength properties and hardness after heat treatment for processing temperatures not exceeding 600 °C. After exceeding this value, the tensile strength and hardness are much lower compared to the base material [20,21]. With this in mind, the use of TCMP steels must go hand in hand with a broad knowledge of their behavior in the production process. In order to control the properties and microstructure of welded joints, it is necessary to conduct extensive research explaining the impact of both the welding process and the applied heat treatment on the structural changes taking place [22,23,24,25]. Due to the interaction of thermal cycles and the resulting stresses, identifying the causes of deterioration of properties is not very easy [23,26,27].

Modern software for simulating welding and heat treatment processes may come in handy in this situation. However, it often happens that the welding technology tested on small samples does not give the same results on large structures where we deal with different heat dissipation or stiffening of elements. In such situations, one solution to realistically determine the distribution and level of stresses in the structure are the mentioned calculation programs. Correctly prepared numerical simulations allow for obtaining a lot of additional information about the welding results, but also about the process [28,29,30]. This allows for the optimization of the process, the detection of potential faults as well as the recent determination of the lifetime of the structure under specific operating conditions. It should be mentioned that of all the numerical analyses carried out using the finite element method (FEM), simulations of welding and heat treatment processes are some of the most complex and difficult types to perform. They require the correct preparation of both the model itself and the material database containing material data in function of the temperature and metallurgical changes taking place in the material, as well as precise calibration and validation of calculation models [31,32].

The main problem of numerical simulations of welding and heat treatment processes is the appropriate methodology for conducting these analyses and the correct definition of the adopted assumptions and material models. A frequent error in calculations, i.e., not considering the structural changes occurring with temperature changes, may result in significant differences in the obtained results compared to reality. Therefore, to maintain the credibility of the results of the numerical analyses, the tool used to carry it out must cover the phenomena that occur during the modeled process as much as possible. The use of random calculation tools which are not specialized unfortunately may lead to incorrect results [31,33,34,35,36].

In the numerical simulations presented in the article, the VisualWeld (SYSWELD) software package (ESI Group, Paris, France) was used. The SYSWELD solver, like many tools for calculating thermal problems in FEM, is based on the calculation of temperature fields based on the Fourier differential formula. However, in SYSWELD it is modified by the shares of metallurgical phases heat convection equation, which, thanks to the coupled thermo-metallurgical analysis, is represented by the relationship [35,36]:(1)$$\left(\underset{i}{\sum }P_{i}{\left(\rho C\right)}_{i}\right)\frac{\partial T}{\partial t}-𝛻\left(\left(\underset{i}{\sum }P_{i}\lambda_{i}\right)𝛻T\right)+\underset{i<j}{\sum }L_{ij}\left(T\right)\cdot A_{ij}=Q$$
where:*T*—temperature [K],*t*—time [s],*x, y, z*—individual coordinates in the axes of the system [m],*a*—thermal diffusivity coefficient [m^2^⋅s^−1^],*λ* —heat conductivity coefficient [W⋅m^−1^ K^−1^],*c*—specific heat [J⋅kg^−1^ K^−1^],*ρ* —mass density [kg⋅m^−3^],*P*—phase proportion,*i,j*—phases index,*Q*—heat source,*L_ij_(T)*—latent heat of *i**→**j* transformation,*A_ij_*—proportion of phase *i* transformed to *j* in time unit.

This allows for a much more precise representation of the material behavior during heating and cooling operations at different speeds of the materials to be welded or processed.

To sum up, the use of modern research techniques in combination with the possibilities offered by modern software for conducting numerical analyses, with a correctly adopted research methodology, allows us to obtain many answers to many questions regarding the behavior of advanced materials in manufacturing processes and to fully use their specific properties. The presented work presents both laboratory results of research on the influence of the annealing temperature of S700MC steel, later confirmed by the results of numerical analyses, as well as several examples of real joints welded using various techniques.

## 2. Description of the Problem

The aim of the experimental part of the research was to determine the relationship between the applied parameters of thermal cycles during welding and heat treatment (annealing) and the values and distributions of residual stresses in the tested samples of S700MC steel. For this purpose, on a test stand equipped with a Gleeble 3500 thermal cycle simulator (Dynamic System Inc., New York, NY, USA), three heat cycles differing in the maximum temperature value were applied on specially prepared samples. Then the residual stresses in the samples were determined by using X-ray diffraction analysis. These samples were annealed in a vacuum furnace for a few selected maximum temperatures ranging from 300 to 650 °C. After the samples had cooled down, the stresses were measured again. As the next step, an attempt was made to perform numerical analyses in order to illustrate the phenomenon under study, and create the basis for the methodology of conducting numerical analyses of the impact of changes in annealing temperature after welding thermal cycle on the values and distribution of residual stresses. Several sample results for various welding methods were also presented.

### 2.1. Material Used for Annealing Tests

The material used for the experiments was S700MC steel, from the group of HSLA (High Strength Low Alloy) steels. It is a structural, fine-grained high-strength steel with a minimum guaranteed yield strength of 700 MPa, which is suitable for cold forming and can be welded. This steel is thermomechanically rolled and microalloyed. The alloys are predominantly aluminum, titanium, niobium and vanadium. Said elements form fine carbides, nitrides or carbonitrides, which contribute to the refinement of the grain and the strengthening of the matrix. The secondary effect of grain refinement is in increased values of yield and tensile strength, as well as in reduced values of transient temperature and brittle fracture properties. The chemical composition of S700MC steel defined by the standard and determined experimentally using a Bruker Q4 TASMAN spectrometer was shown in Table 1.

The measurements on the spectrometer showed that the tested steel has a very specifically low carbon content (0.056%), therefore the strength and plastic properties of welded joints may be reduced as a result of limiting the influence of austenite changes during the welding process. Because of this, it was also necessary to modify the SYSWELD material base used in the performed numerical analyses to compensate for the differences in the chemical composition of the existing base and the base needed for calculations. The mechanical properties were determined experimentally by means of a static tensile test and they are presented in Table 2.

### 2.2. Assumptions for Testing Stresses Distribution in S700MC Steel under the Conditions of Welding and Annealing Process Thermal Cycles

In order to determine the influence of the annealing temperature on the magnitude and stresses distribution after the welding process, tests were carried out on the Gleeble 3500 thermal cycle simulator. The experiment was planned in such a way that it was possible to determine both the influence of the simulated thermal cycle of the welding process and the post-welding heat treatment thermal cycle. For this purpose, samples with dimensions 10 × 10 × 102 mm were prepared. An M10 thread was cut at the ends of the specimens to secure the specimen in the jaws of the instrument. The square cross-section was selected so that when measuring the residual stresses using the X-ray diffraction method (XRD), there was no need to compensate for the shape of the sample surface, which affects the accuracy of the measurement. For the finished samples, it was necessary to remove the undesirable residual stresses in the surface layer created after machining. For first group of samples, this was achieved by heating in a vacuum furnace at a temperature of 650 °C for 2 h, which are annealing parameters commonly used in technical practice, in the heat treatment of welded assemblies. In the second group of samples, the surface layer was removed by local electrolytic etching. For this case, a special silicone nozzle with dimensions 7 × 15 mm was made with the help of 3D printing, which copied the surface of the sample and thus resulted in uniform etching. The thickness of the etched layer was 0.2 mm. The sample with the etched surface layer is shown in Figure 1a.

The samples were then prepared for tests on the Gleeble 3500 simulator. First, a K-type thermocouple was welded to the center of each sample to control the heating and cooling temperature cycles. After welding the thermocouple, two nuts were screwed onto both ends of the sample, firstly to ensure proper clamping in the temperature simulator, but mainly so that the sample could be exposed to tensile and compressive stresses. Finally, the sample was inserted into a copper jaws with full contact, as shown in Figure 1b, so that a temperature gradient corresponding to the conditions during real welding is created in the sample. For this reason, the free length of the sample between the jaws was limited to 10 mm. The whole assembly was inserted into the working chamber of the Gleeble 3500 simulator. For correct and sufficient contact between the cooper jaws and the clamping system of the device, the sample was fixed with spacer bolts. The primary reason for this step is to ensure good current transfer heating the sample and to ensure intensive heat dissipation through the copper jaws. A secondary reason for the above procedure is to secure the specimen against movement so that tensile and compressive stresses can be applied to it. The clamped specimen located in the working chamber of the Gleeble system is shown in Figure 2.

The Gleeble 3500 system allows heating at speeds of up to 10,000 °C/s. Therefore, real welding cycles measured by arc welding thermocouples can be applied to the sample. Figure 3 shows the temperature cycle used, which was recorded during GMA welding of a S700MC steel fillet weld. Based on it, three temperature cycles with different maximum temperatures were programmed: 1080, 1200 and 1365 °C. Thus, the temperature cycles corresponding to the three different locations in the Heat Affected Zone (HAZ).

The next step was to set the method of program control. The Gleeble system allows several ways to control the process, with power control being the most commonly used method. In such a case, the control system makes it possible, thanks to the feedback control and the load cell, to apply a defined force (or stress) or a zero force to the samples during heating and cooling. In this case, the jaws are moved according to the data from the load cell so that no residual stresses or plastic deformations occur in the sample. In welding simulations, control by means of a strain gauge is usually used, when a defined displacement of the specimen or zero elongation of the specimen can be realized. This simulates the clamping stiffness of the specimen, as in the real welding case. As the thickness of the welded materials increases, the intrinsic stiffness of the entire system also increases. Therefore, a welding method using a strain gauge is used in welding simulations. Then it is possible to define how much the sample can be extended during heating, or with zero possibility of elongation it is possible to define a perfectly rigid clamping of the sample. This method was chosen when applying the programmed temperature cycles to the test specimens. In addition, it was tested whether and, if so, how the increase in cooling rate to the value of residual stresses in the sample would occur. Therefore, both temperature cycles corresponding to reality and temperature cycles with so-called free cooling were used, where after reaching the maximum temperature the controlled reheating is switched off and the sample is cooled only by heat dissipation through high temperature jaws with full contact. In this case, the sample was cooled approximately 2.5 times faster than controlled cooling. Both used thermal cycles with different cooling methods were shown in Figure 3. The free cooling curve shows a delay at about 480 °C, caused by a transformation that releases latent heat.

After programming the temperature cycle and evacuating the working chamber to a value of 0.533 Pa (4·10^−3^ Torr), the test itself could be started. For all samples, the preparation and test were repeated in the same way, only with the difference between the set maximum cycle temperature and the above-mentioned cooling method. During the experiments, the temperature, forces and stresses in the sample were measured and the possible elongation of the sample was also checked. Figure 4 shows the dependence of the stresses generated in the sample during heating to the maximum temperature of the cycle.

However, it should be noted that the temperature in the sample was not uniform (this was a temperature gradient), and therefore each part of the sample behaves differently, as in HAZ welds. It is also clear from the graph at Figure 4 that after the start of heating in the sample, the compressive stresses first increase, and only reach the yield point at a temperature of about 300 °C (red line at Figure 4). Furthermore, the sample was plastically deformed, so the stress value stabilized at around 400 MPa and the temperature continued to rise. After reaching a temperature of approx. 750 °C, there was a significant decrease in compressive stresses, which was caused by the transformational conversion of the material to austenite. After reaching the maximum cycle temperature and the start of cooling (blue line at Figure 4), the compressive stresses were converted into tensile stresses, the value of which increased during cooling. When the transformation temperature of austenite was reached, the stress was further significantly reduced. With further cooling, the tensile stresses continue to increase, as can be seen in Figure 4.

## 3. Results of the Tests on the Thermal Cycles Simulator

The experiment plan consisted of the following points:(1)use of welding thermal cycles at Gleeble 3500 simulator with a maximum cycle temperature of 1365, 1200 and 1080 °C and a cooling rate corresponding to the measured welding cycle,(2)application of the above thermal cycles at Gleeble 3500 simulator with controlled cooling rate,(3)determination of the conditions for lowering the value of residual stresses and quantifying the values for individual temperatures and strength at these temperatures. It was decided to perform the experiments at the temperatures 200, 300, 400, 500 and 550 °C with the hold time at 2 and 4 h,(4)determination of mechanical properties of S700MC steel after annealing in selected process conditions.

### 3.1. Residual Stresses Distribution after Welding Thermal Cycle Application

After applying the programmed temperature cycles to all tested samples, it was possible to proceed to the measurement of internal residual stresses. The measurement was performed by X-ray diffraction (XRD) analysis using a Proto iXRD combo instrument (Proto Manufacturing Inc., Taylor, MI, USA) with a chromium X-ray tube, as can be seen in Figure 5.

#### 3.1.1. Samples with a Maximum Cycle Temperature at 1365 °C

In the case of testing samples subjected to the impact of the welding thermal cycle with a maximum temperature of 1365 °C, the following samples were made:samples no. 1 and 4—these were annealed to reduce residual stresses at temperature 650 °C for 2 h. This was followed by a thermal cycle with a maximum temperature of 1365 °C and free cooling.sample no. 3—this was annealed to reduce residual stresses at temperature 650 °C for 2 h. This was followed by a thermal cycle with a maximum temperature of 1365 °C with controlled cooling.sample no. 5—a 0.2 mm surface layer was etched away. This was followed by a thermal cycle with a maximum temperature of 1365 °C with controlled cooling without annealing at the beginning.

The achieved stresses profiles for samples no. 1 and 4 are shown in Figure 6.

The reason why two tests were performed under completely identical conditions was to assess the repeatability of the experiments, and thus the relevance of the obtained results. It can be seen from the Figure 6 that the measured stresses distributions and values were almost identical in these two samples. Thus, the repeatability of the measurements for the same boundary conditions of the experiment can be assumed. Therefore, and also due to the high time and cost of each experiment, measurements were performed for only one sample at a time. The green line in Figure 6 indicates the initial stresses state in the sample after annealing to reduce the residual stresses (650 °C for 2 h) and before testing thermal cycle application. The stresses on sample no. 3 with controlled cooling and sample no. 1, which was freely cooled, were also compared, see Figure 6.

Stresses distribution on sample no. 5 where the thermal cycle with a maximum temperature of 1365 °C was applied with no initial annealing was performed in comparison with sample no. 3 were shown on Figure 7. In such a sample, high values of residual stresses reaching a depth of about 0.1 mm are introduced into the surface layer after machining. Therefore, the affected surface layer was etched to a depth of 0.2 mm so that only the unaffected material was measured. In order to remove the thin surface layer, closed circuit electrolytic etching with the Electrolyte type A supplied by Proto Manufacturing Europe was used. Due to the constant flow of electrolyte, the etched area was not thermally affected. The internal stresses after the welding thermal cycle simulation were measured in only one half of the sample. The reason was to maintain the same boundary conditions of the experiment, as the size of the etched area was limited by the dimensions of the etching nozzle to 7 × 15 mm. From previous measurements, the symmetry of the distribution of internal stresses in the right and left halves from the center of the sample was assumed. In sample no. 5, in comparison to the previous samples, the internal stresses were further measured into the base material so as to reach a steady state. In the case of both samples, it was a simulation with controlled cooling.

#### 3.1.2. Samples with a Maximum Cycle Temperature at 1200 °C

In the case of samples heated by the welding thermal cycle with a maximum temperature of 1200 °C, the following samples were tested:sample no. 6—heated with a welding thermal cycle to a maximum temperature of 1200 °C with controlled cooling,sample no. 7—heated with a welding thermal cycle to a maximum temperature of 1200 °C with free cooling.

Both samples had an etched surface layer to a depth of 0.2 mm. As with sample no. 5, the residual stresses were measured on only one half of the sample, to maintain the same boundary conditions of the experiment, as can be seen in Figure 8.

#### 3.1.3. Samples with a Maximum Cycle Temperature at 1080 °C

The last experiment on Gleeble was the application of a welding thermal cycle with a maximum temperature of 1080 °C. In this case, only one type of experiment was performed as sample no. 2. This sample was annealed to reduce residual stresses at 650 °C for 2 h. This was followed by a thermal cycle with a maximum temperature of 1080 °C with controlled cooling. The course of the residual stresses across the sample after the application of this cycle was significantly different from the cycles with a maximum temperature of 1365 and 1200 °C. The curve in the center of the sample did not tend to decrease significantly, as can be seen in Figure 9.

### 3.2. Influence of Annealing on the Change of the Magnitude of Residual Stresses

To assess the effect of annealing on the magnitude of residual stresses in the material, two annealing diagrams were experimentally compiled. One lasts 2 h and the other lasts 4 h at the annealing temperature. The samples used in this experiment were annealed in a vacuum furnace at 200, 300, 400, 500 and 550 °C. After annealing all samples, the residual stresses values were measured by X-ray diffraction analysis. In order to measure the initial stress of the base material, a part of the material was etched from the surface of the sample to a depth of 0.2 mm. The measured values of residual stresses after annealing with endurance at temperature for 2 and 4 h were summarized in Figure 10 and Table 3. Table 3, on the one hand, indicates the initial state of stress at a given point after thermomechanical rolling of the S700MC steel, and then the subsequent value of the residual stress after the selected annealing temperature cycle to reduce the residual stresses. To measure the initial stress of the base material, a part of the material was etched from the surface of the sample to a depth of 0.2 mm.

### 3.3. Influence of Annealing on the Reduction Of Residual Stresses after Temperature Cycles

To assess the influence of the annealing temperature on the residual stresses value results after welding thermal cycle sample no. 1 was tested as an example. First the specimen was annealed at 650 °C for 2 h before heating through the welding thermal cycle. Then the welding cycle with a maximum temperature of 1365 °C was applied to the sample. After applying the welding thermal cycle and measuring the internal stresses, annealing was performed to reduce the internal stresses level. For this case, annealing at 300 °C for 2 h was chosen. After annealing the internal stresses was again measured at the same points by X-ray diffraction analysis, Figure 11. It should be noted here that after annealing, the compressive stresses almost disappeared and the tensile stresses decreased, of course, considering the dependence that the higher the annealing temperature, the lower the residual stresses values. This relationship was also shown later in the numerical simulation results.

### 3.4. Determination of Mechanical Properties after Annealing

For completeness of measurements, a static tensile test was performed on selected annealed samples. From this, the mechanical properties of the heat-treated samples were determined and compared with the mechanical properties of the base unannealed material, which were listed in Table 2. First, two samples annealed at 450 °C for 2 h were measured. Subsequently, a tensile test was performed on two samples annealed at 550 °C for 4 h, as can be seen in Table 4.

## 4. Numerical Simulations

While on small samples or test joints, the laboratory tests of the influence of the annealing temperature on the mechanical properties and stresses distribution do not pose any major problems except to the measurement accuracy and by causing possible disturbances in the measured values, though this is problematic in the case of large structures and ready-made implementations. The fact that it is difficult or impossible to carry out XRD tests and impossible to remove the surface layer under these circumstances, means that modern techniques used in welding, such as numerical analyses, can help. For this purpose, four examples related to the issue of annealing effect on stresses distributions and values are presented later in this article. All analyses presented were calculated in VisualWeld (code SYSWELD). The S700MC steel material database used in the simulations contains the full thermal, metallurgical and mechanical material properties depended on the temperature and metallurgical phases changes and was modified according to measured chemical composition, as can be seen in Table 1.

### 4.1. An Example of the Numerical Model of Tests Carried Out on the Simulator of Thermal Cycles

The demonstration of the usefulness of the tool, which are numerical analyses of welding and heat treatment processes, began with an attempt to simulate the tests carried out on a thermal cycle simulator. For this purpose, a two-dimensional numerical model was created containing only 800 2D elements and 861 nodes, as can be seen in Figure 12a. The model mesh was refined in the place of the defined thermal cycle. A model of this type drastically reduces the computation time and, in some cases, allows for almost immediate results to be obtained. This model was developed as a 2D-symetrical model and the calculations were performed using the ‘transient’ (step-by-step) technique with the Imposed Thermal Cycle load [31]. This method relies on loading selected mesh nodes with a defined heat cycle instead of using the frequently used moving heat source model, as can be seen in Figure 12b.

In the case of using this computational technique, it is essential to correctly calibrate the model and reproduce the course of the heating curve, and then determine the boundary conditions so that the temperature values and cooling speed obtained in the simulation correspond to those obtained in real tests during thermoelectrical measurements. After the initial calibration of the numerical model, analyses were carried out for three selected thermal cycles simulating the influence of the welding process with maximum temperatures: 1360, 1200 and 1080 °C, as can be seen in Figure 13a. According to the methodology of calculating with the use of the predefined thermal cycle, the selected mesh elements were loaded with the heating curve until the maximum temperature was reached, and then the cooling proceeded according to the assumed boundary conditions. The boundary conditions corresponding to the clamping were set to simulate heating and cooling of the elements corresponding to the conditions of mounting the real sample in the jaws of the Gleeble simulator. Additionally, the symmetry boundary condition was used due to the fact that the computational model was a symmetrical model, as can be seen in Figure 12a. The conditions for heat dissipation were determined by such heat dissipation at the external nodes of the model mesh (except for the nodes marked as the axis of symmetry) that in the welding cycle simulation phase, the calculated cycle was consistent with the one recorded on the simulator and then in the phase heat treatment according to the given thermal cycle of annealing, as can be seen in Figure 13. The calculated distribution of average stresses for the selected thermal cycles do not differ noticeably in terms of the obtained values of the calculated stresses. However, it should be noted that their distributions differ slightly, as can be seen in Figure 14.

In the next step, numerical analyses of the annealing process were carried out at 400, 550 and 650 °C withstanding the element at these temperatures for 2 h in accordance with the cycle shown in Figure 13b. The samples were heated on all mesh nodes at the rate of 5 °C/min and cooled to the ambient temperature at the rate of 3 °C/min. The stresses distribution and graphs of their values on the cross-section in the middle of the sample thickness for selected thermal cycles and temperatures of the annealing process were shown in Figure 15 and Figure 16. There is a visible decrease in the value of stresses with the increase of the annealing temperature and at the temperature value of approximately 650 °C, the stresses value does not exceed the value of 50 MPa, as can be seen in Figure 16 and Figure 17. The calculated values and their course are also similar in some areas to the values recorded during the actual tests carried out on the thermal cycle simulator, Figure 6, Figure 7, Figure 10 and Figure 11. In the real tests, for the cycle with a maximum temperature of 1080 degrees, it was not measurements for the etched samples. Both for cycle 1360 were provided, i.e., for the sample with and without the surface layer removed. The FEM simulation does not show the peak compressive stresses because the grinding of the sample during its preparation was not simulated. Consequently, the simulation results correspond to the results obtained for the etched samples.

### 4.2. An Example of a Numerical Analysis of a Multipass Butt Joint Gas Metal Arc Welding and Annealing Process

After numerical analyses of laboratory tests performed on a thermal cycle simulator, the analysis of a model butt joint, welded with two beads, was carried out on a 2D cross-section model using the ‘transient’ technique [31]. The numerical model prepared for the Gas Metal Arc Welding (GMAW) welding of two 10 mm thick plates contained 552 elements (440 2D elements and 112 elements of 1D type) and 485 nodes, as can be seen in Figure 18.

The boundary conditions describing the clamping of plates to be welded have been adopted in such a way as to simulate the welding situation without additional fastening. The heat transfer from the model to environment was defined at the outer edges of the model as convective heat dissipation and radiation to the environment at 20 °C.

Heat source models in VisualWeld (SYSWELD) are described by a volume density of energy applied to elements Q(x,y,z) [31,32,35]. In the described case as a model of a moving heat source in the ‘transient’ technique (step-by-step), a double ellipsoid volumetric model (Goldak’s model) was used, as can be seen in Figure 18 and Figure 19.

In SYSWELD, it is possible to introduce the power density function applied to the structure. Accordingly, the energy was divided into:

*Q_f_*—as the heat energy density introduced in the front half of the model
(2)$$Q_{f}\left(x,y,z\right)=\frac{6\sqrt{3}f_{f}Q}{abc_{f}\pi \sqrt{\pi }}exp\left(\frac{-3x^{2}}{a^{2}}\right)exp\left(\frac{-3y^{2}}{b^{2}}\right)exp\left(\frac{-3z^{2}}{c^{2}}\right)$$

*Q_r_*—as the heat energy density introduced in the rear part:(3)$$Q_{r}\left(x,y,z\right)=\frac{6\sqrt{3}f_{r}Q}{abc_{r}\pi \sqrt{\pi }}exp\left(\frac{-3x^{2}}{a^{2}}\right)exp\left(\frac{-3y^{2}}{b^{2}}\right)exp\left(\frac{-3z^{2}}{c^{2}}\right)$$
where *Q* is the total power source; *a*, *b*, *c_f_/c_r_* are respectively width/depth/length of the front/rear part of the molten pool; *f_f_*, *f_r_* are constants that affect the flow of energy in a material. It must be also mentioned that *f_f_* + *f_r_* = 2 [31,33].

After calibration procedure of the heat source models, the optimal parameters were determined, as can be seen in Figure 20 and Table 5. Then the mechanical analysis for welding process was carried out, as can be seen in Figure 21.

After simulation of welding, as in the previous described case, numerical analyses of the annealing process at temperatures of 400, 550 and 650 °C were carried out in accordance with the thermal cycles shown in Figure 13b. The welded joint was heated in all mesh nodes at the rate of 5 °C/min, withstood at a given temperature for 2 h and then cooled at the rate of 3 °C/min to the ambient temperature. The obtained stresses distributions and graphs of their values analyzed in three parallel lines along the cross-section of the model were shown in Figure 22 and Figure 23. There is observed a noticeable decrease in the value of the calculated stresses, while maintaining their type (compressive/tensile) with an increase in the annealing temperature. In the case of annealing temperature of 650 °C, a decrease in the calculated stresses to values not exceeding approximately 30 MPa was visible, as can be seen in Figure 22b and Figure 23.

### 4.3. An Example of a Numerical Analysis of the Laser Butt Joint Welding and Annealing Process

As a further example, a symmetrical three-dimensional numerical model of the of 10 mm thick laser-welded butt joint with filler material was prepared. This model consisted of 32,091 3D solid elements with 29,987 nodes. As in the previous case, the boundary conditions related to heat dissipation to the environment were defined on all external surfaces of the model as convective heat dissipation and radiation to the environment at 25 °C. The boundary conditions related to the fixing of the welded elements were set in such a way that the model behaves like a sample lying on the welding table without any additional fixing. The additional boundary condition of symmetry was used on the model split plane because the computational model was a symmetrical model. The analyses were carried out using the ‘transient’ technique with a moving model of the heat source in the form of 3D conical which can be determined by the defined power and its dimensions, as can be seen in Figure 24.

The mathematical description of this model is contained in two equations which describes [31,33]:

heat transfer to the material structure depending on the given coordinates
(4)$$Q\left(x,y,z\right)=Q_{0}exp\left(-\frac{x^{2}+y^{2}}{r_{0}^{2}\left(z\right)}\right)$$
and the change in radius at depth
(5)$$r_{0}^{2}\left(z\right)=r_{e}+\frac{r_{i}-r_{e}}{z_{i}-z_{e}}\left(z-z_{e}\right)$$
where: *Q*_0_ is the heat flux density, *r_e_*, *r_i,_ z_e_*, *z_i_* are the cone geometrical dimensions according to Figure 24.

The calculations were performed using the “transient” calculation technique with a moving heat source model. After calibrating procedure of the heat source model and obtaining full remelting of the welded sheets, a mechanical analysis was also carried out to determine the stresses distribution in the analyzed joint after welding process, as can be seen in Figure 25, Table 6.

After welding and cooling the sample, a thermal cycle was set for all external nodes of the model corresponding to annealing at 400 and 650 °C, Figure 26 and Figure 27. The calculated stresses distributions were analyzed in the same way as before in three parallel lines drawn on the cross-section of the model, as shown in the Figure 28 with graphs of the stress distribution in individual lines.

### 4.4. An Example of a Numerical Analysis of a Hybrid Butt Joint Welding and Annealing Process

The last of the presented examples was the complex process of hybrid welding. As a combination of two welding processes: laser and arc welding, the method allows for high efficiency on the one hand and, on the other hand, significantly reduces the risk of high stresses, as is often the case with laser welding. For this purpose, a discrete full three-dimensional model of a hybrid-welded butt joint was prepared, consisting of 45,748 3D solid elements with 41,321 nodes. The mesh was concentrated in the weld area to increase calculation accuracy, as can be seen in Figure 29. Boundary conditions related to clamping conditions during welding were set to simulate welding without any additional fastening. For boundary conditions corresponding to heat dissipation to the environment, it was assumed that on each external surface, convective heat dissipation occurred to the environment at 20 °C and the radiation point.

In the case of numerical analyses of the hybrid welding process, a double ellipsoid-shaped heat source model to model the arc welding process and a 3D conical model for laser welding modeling were used, as can be seen in Figure 29. The calculations, as in the previous case, were performed using the “transient” technique. In order to calibrate the heat source models, an in-built “Heat Input Fitting” module was used to optimize the shape of the resulting molten metal pool, Figure 30. Additionally, it should be noted that, in order to be fully consistent with reality, the Goldak model followed the cone model at a distance of 4.0 mm. The results of the calibration were presented in Figure 30, Table 7.

The obtained stresses distributions after the welding process and the sample cooling down correspond to the heat distribution during welding and the maximum stresses values are concentrated in HAZ, Figure 31. As occurred previously, numerical analyses of the annealing process were carried out at temperatures of 400, 550 and 650 °C in accordance with the cycles shown in Figure 13b.

Also, in this case, the calculated stresses distributions in the analyzed joints behave as in the cases described earlier. Graphs of the stresses values in three parallel lines, drawn in a cross-section on the surface, in the middle of the joint and on the bottom, also showed that an increase in the annealing temperature causes a decrease in the stress value while maintaining their character. In the case of annealing at the temperature of 400 °C, only a slight decrease in the value is visible, which in the case of increasing the temperature by another 150 °C is more noticeable—especially at the bottom of the weld joint, as can be seen in Figure 32 and Figure 33.

Annealing at the temperature of 650 °C, similar to the previous analyzed cases, resulted in a decrease in the stresses values in the weld area and HAZ to the maximum value of 53 MPa and in the further parts of the joint practically up to a few MPa, as can be seen in Figure 32b and Figure 33.

## 5. Conclusions

The use of modern construction materials, which often obtain their high strength properties as a result of complex thermo-mechanical treatment, often creates problems of a technological nature. On the one hand, the thermal cycle of the welding process, and on the other hand, post-welding heat treatment used to reduce and homogenize the stresses distribution, often cause changes in the precisely obtained, fine-grained structure as well as in strength and plastic properties. The temperature and duration of the post-weld heat treatment are usually both the cost of this operation and the impact on the recall of material properties. The precise determination of these values is only simple for homogeneous laboratory samples. In the case of real constructions, it is virtually impossible to do precisely. In such situations, numerical analyses of welding processes come in handy.

The test results showed that the essentially similar in nature stresses distributions in the tested samples differ depending on the maximum cycle temperatures in the sizes of max. achieved tensile stresses even by about 250 MPa in the case of the cycle with temperature 1360 and 1080 °C, Figure 6 and Figure 9. It should also be noted that in the case of max. values of compressive stresses in the sample for each of the tested cycles, this value is similar and oscillates at around 500–550 MPa, as can be seen in Figure 6, Figure 7 and Figure 9. Such high values were caused by the process of grinding samples at the stage of their preparation. Removal of a small (0.2 mm thick) top layer of the sample caused a decrease of this value to about 100 MPa, Figure 7 and Figure 9. Of course, it should be mentioned that etching does not have the same effect as annealing. During annealing, residual stresses in the entire cross-section of the sample are minimized, not only in the surface layer affected by machining. Only the affected surface layer is removed by etching, but the residual stresses after thermomechanical processing remain in the entire cross section of the sample.

For the cycles with slow and controlled (accelerated) cooling, some differences are also visible in the stresses diagrams. For a cycle with a maximum temperature of 1365 °C, it is visible that the stresses in the case of forced cooling obtain values much lower, even by 150 MPa (approximately 400 MPa in relation to almost 500–550 MPa), as well as in the center of the sample, their significant decrease in relation to for the case of slow cooling, as can be seen in Figure 6 and Figure 8. The most likely reason for is a longer time for compensation of residual stresses by plastic deformation and a slightly longer time for stresses relaxation. The annealing tests carried out in the temperature range from 300 to 550 °C showed that the value of stresses in the workpiece decreased with the temperature increase, as can be seen in Figure 10 and Table 3. In addition, on the basis of the research, it was also found that the extension of the holding time at the given temperature of the heat treatment from 2 to 4 h causes a decrease in the final value of the measured stresses from approx. 60 to 4–9 MPa for the annealing temperature equal to 550 °C, as can be seen in Table 3 and Figure 10. Transferring this experiment to a sample subjected to the effect of the welding thermal cycle confirmed the decrease in the stress level to the value of 250 MPa, which seems to be close to the previously created annealing diagram, as can be seen in Figs. 10 and 11. However, it should be also noted that both the increase in the temperature of the treatment and the extension of its duration cause a decrease in the tensile strength limit by an average of 20–40 MPa with almost constant plastic properties, as can be seen in Table 5. It may therefore be caused by a partial, slight stress relaxation at these temperatures, which, however, does not remove the strengthening mechanisms of the material resulting from its production process.

Numerical analyses of the samples processed in the thermal simulator showed a high correlation of the results with the actual tests. The differences between the actual and calculated values should be explained on the one hand by the simplifications of the numerical model (defining of boundary conditions, accuracy of the prepared material base, etc.), and on the other hand also by possible measurement errors and disturbances, which may also occur in the case of real test. Nevertheless, a significant convergence of the simulation results with reality is observed, which in the case of purely engineering applications is more than acceptable, as can be seen in Figure 6, Figure 10, Figure 11, Figure 14, Figure 15, Figure 16 and Figure 17.

The distributions and values of the calculated stresses were consistent with the results obtained for samples subjected to electrolytic etching. This is understandable due to the fact that this operation has not been modeled. However, it requires adopting an assumption in the methodology of conducting this type of research that the measurements of stresses in real samples should be performed after removing the effects of the stage of sample preparation for testing or possibly take into account the presence of a thin layer of high compressive stresses. Based on the proven methodology, several examples of numerical analyses carried out for various welding processes as well as various calculation techniques used in modern programs for the analysis of welding and heat treatment processes are presented. In the presented models, both full three-dimensional models and techniques with a very reduced time needed to perform the calculations were used to answer the key question: whether and if the levels of residual stresses for the selected heat treatment variant will be lowered. In each of the analyzed cases, as a result of the thermal cycle of the annealing process, the value of the calculated stresses decreases, and after exceeding the temperature of 650 °C, the stress value was practically completely reduced to values not exceeding 30–50 MPa, as can be seen in Figure 16 and Figure 17.

The result of the GMAW multipass calculations were the stresses distributions, which show that with the increase of the annealing temperature, the stresses level in the joint decreases, as can be seen in Figure 23. When processed at the temperature of 650 °C, the residual stresses values drop practically to a level not exceeding 30 MPa. In addition, the use of the 2D cross section technique when building the model allows for quick calculation of many variants that the user considers when developing the necessary technology. In case of numerical analyses of the laser welding process, the calculated stresses after welding reach their maximum values at the level of 544 MPa in the HAZ area, as can be seen in Figure 26. These values are higher than those obtained in the case of arc welding (a maximum of about 300 MPa) and characteristic for laser welding, and cover a much narrower zone. Also, in this case, the applied heat treatment shows a decrease in the value of residual stresses with increasing temperature of the applied annealing process. However, in this case, visible changes are only observed after exceeding the annealing temperature of 550 °C, as can be seen in Figure 27 and Figure 28. The results of the calculations carried out for hybrid welding were a kind of confirmation of the above observations. The use of an arc heat source in hybrid welding “softens” the impact of the acute thermal cycle of the laser beam, as shown by the obtained stress distributions on the cross-section of the joint after the welding process, laser welding, though this only happens in practice after exceeding the annealing temperature of 550 °C, as can be seen in Figure 33.

To sum up, the comparison of the results of the tests and numerical analyses shows that the increase in the annealing temperature in each of the analyzed cases causes a decrease in the level of residual stresses and their distribution. This information should be considered during preparation of the welding procedure and when creating a Welding Procedure Specification (WPS). It is also visible in this case how useful the results can be of a correctly performed numerical analysis of the welding process at this stage, which greatly facilitates the selection of appropriate conditions for carrying out these types of operations. As mentioned earlier, this will be especially important in the case of large, complex and responsible structures.

## Figures and Tables

**Figure 1 materials-13-05289-f001:**
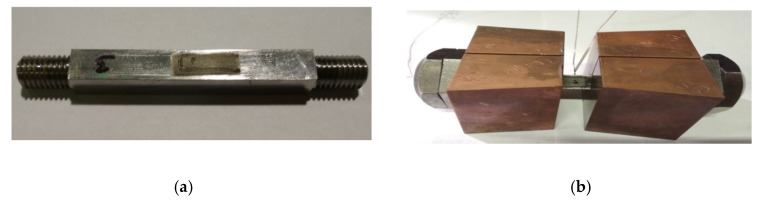
A view of (**a**) test sample with etched area for measurements and (**b**) a sample attached to the tooling with a welded thermocouple.

**Figure 2 materials-13-05289-f002:**
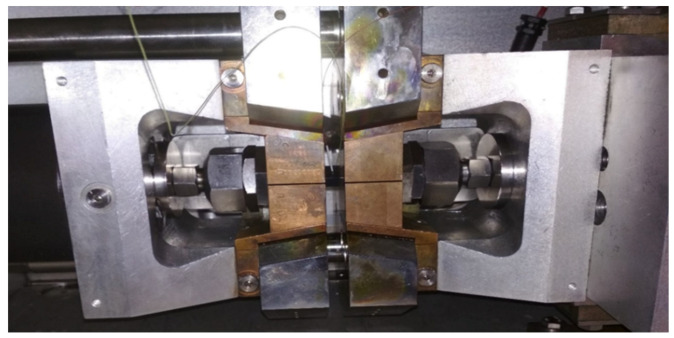
A view of a Gleeble 3500 simulator working chamber with sample attached to the tooling equipment.

**Figure 3 materials-13-05289-f003:**
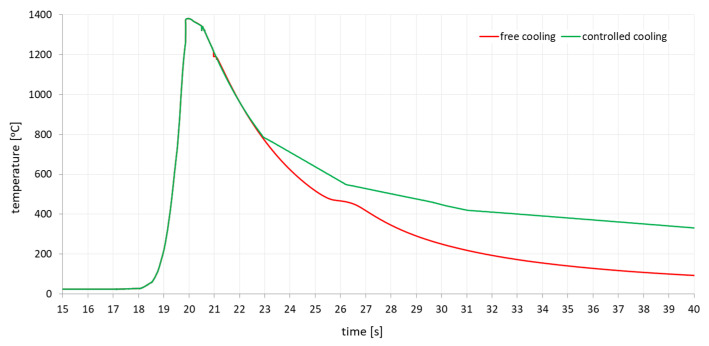
Comparison of controlled cooling thermal cycles (green curve) which simulate welding cycle with thickness of 10 mm and thermal cycle with higher cooling rate (red curve) which simulate welding cycle with thickness higher than 10 mm.

**Figure 4 materials-13-05289-f004:**
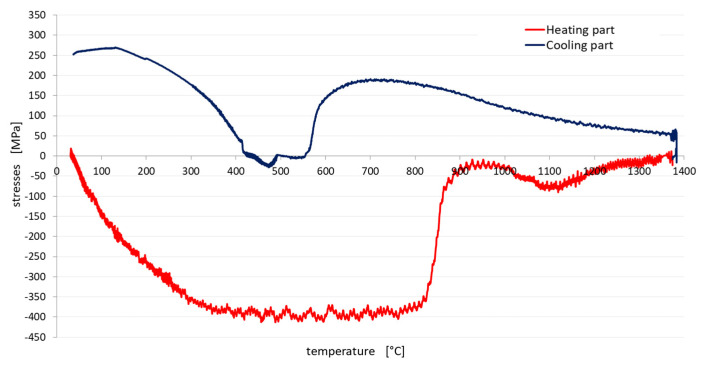
Graph of the dependence of the sample stresses on the temperature during heating and cooling.

**Figure 5 materials-13-05289-f005:**
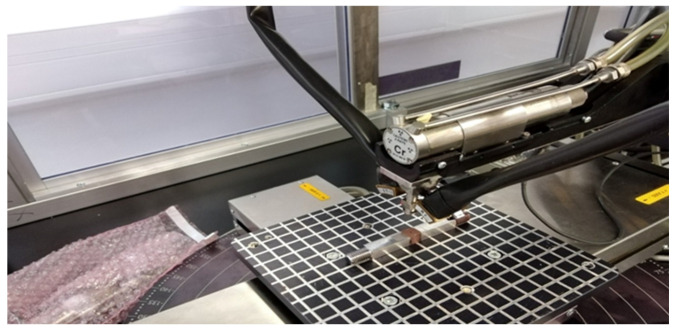
A view of a X-ray diffraction testing stand.

**Figure 6 materials-13-05289-f006:**
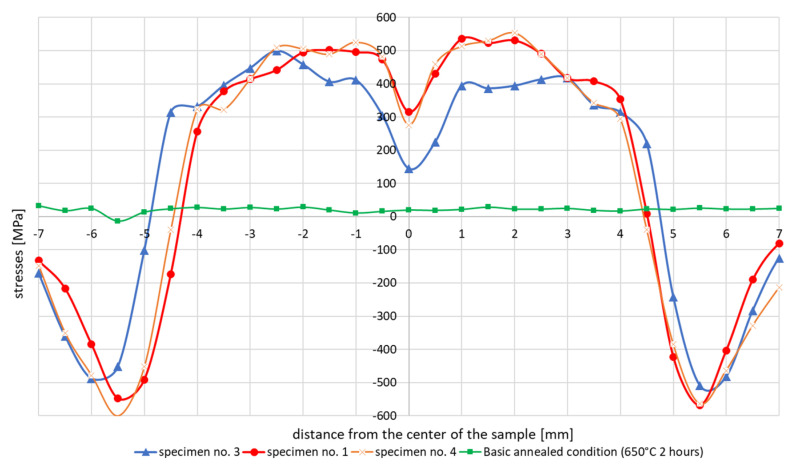
Comparison of the residual stresses distribution in samples with free (specimen 1 and 4) and controlled cooling (specimen 3)—welding thermal cycle with a maximum temperature of 1365 °C.

**Figure 7 materials-13-05289-f007:**
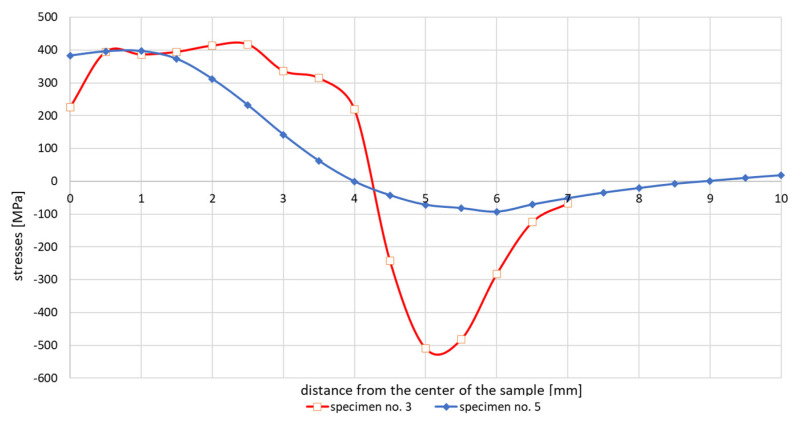
Comparison of residual stresses distribution in etched and not annealed samples with a non-etched sample after a welding cycle with maximum temperature of 1365 °C and controlled cooling.

**Figure 8 materials-13-05289-f008:**
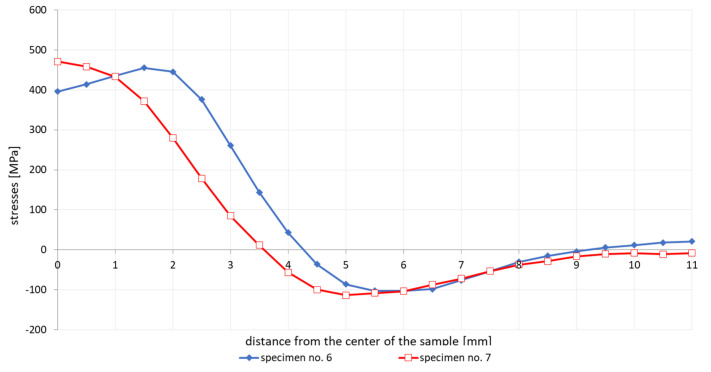
Comparison of the residual stresses distribution in etched samples after a welding cycle with a maximum temperature of 1200 °C and free and controlled cooling.

**Figure 9 materials-13-05289-f009:**
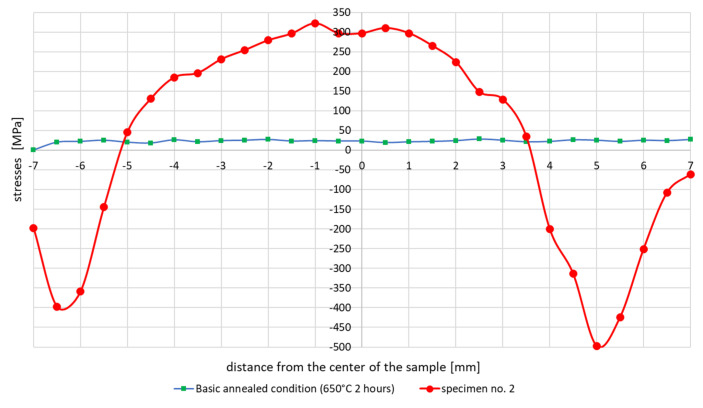
Course of residual stresses in sample 2 after a welding thermal cycle with a maximum temperature of 1080 °C and controlled cooling.

**Figure 10 materials-13-05289-f010:**
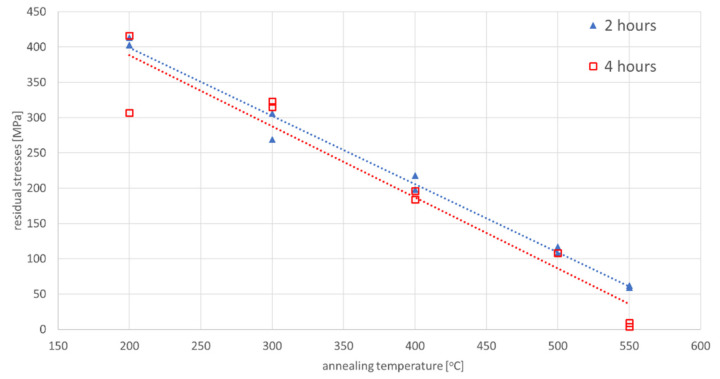
Annealing diagram for S700MC steel for annealing times of 2 h and 4 h.

**Figure 11 materials-13-05289-f011:**
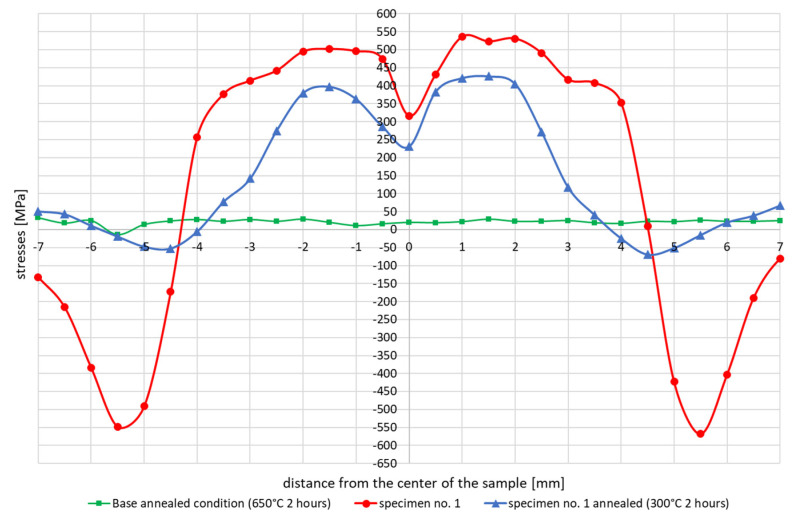
Comparison of the residual stresses distribution in samples subjected to the welding thermal cycle with a maximum temperature of 1365 °C with free cooling that was” annealed at 300 °C for 2 h.

**Figure 12 materials-13-05289-f012:**
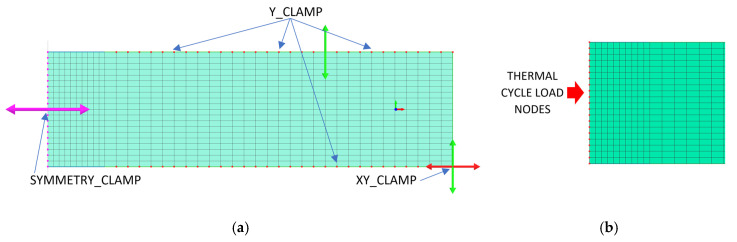
A view of a 2D numerical model with (**a**) clamping conditions and (**b**) “load” nodes area.

**Figure 13 materials-13-05289-f013:**
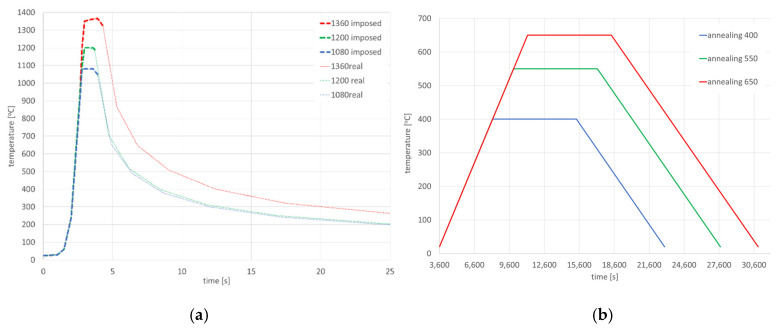
Imposed thermal cycle of (**a**) heating for welding conditions (dashed line is applied cycle and dotted is calculated) and (**b**) thermal cycle of post weld heat treatment (annealing with different temperature).

**Figure 14 materials-13-05289-f014:**
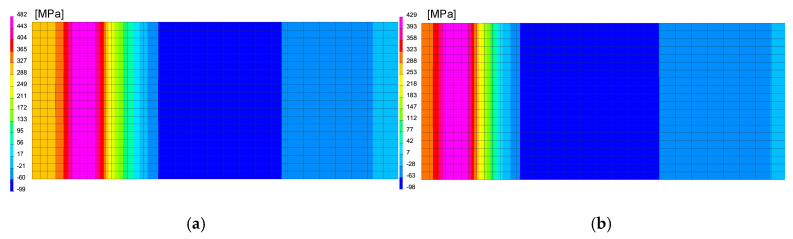
Comparison of the residual mean stresses distribution after welding thermal cycle application for (**a**) 1360 °C and (**b**) 1080 °C thermal cycles.

**Figure 15 materials-13-05289-f015:**
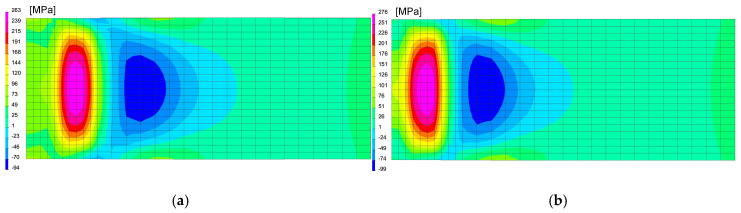
Comparison of the residual mean stresses distribution after annealing in 400 °C by 2 h for (**a**) 1360 °C and (**b**) 1080 °C thermal cycles.

**Figure 16 materials-13-05289-f016:**
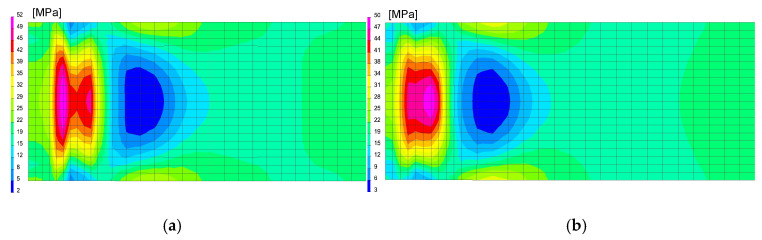
Comparison of the residual mean stresses distribution after annealing in 650 °C by 2 h for (**a**) 1360 °C and (**b**) 1080 °C thermal cycles.

**Figure 17 materials-13-05289-f017:**
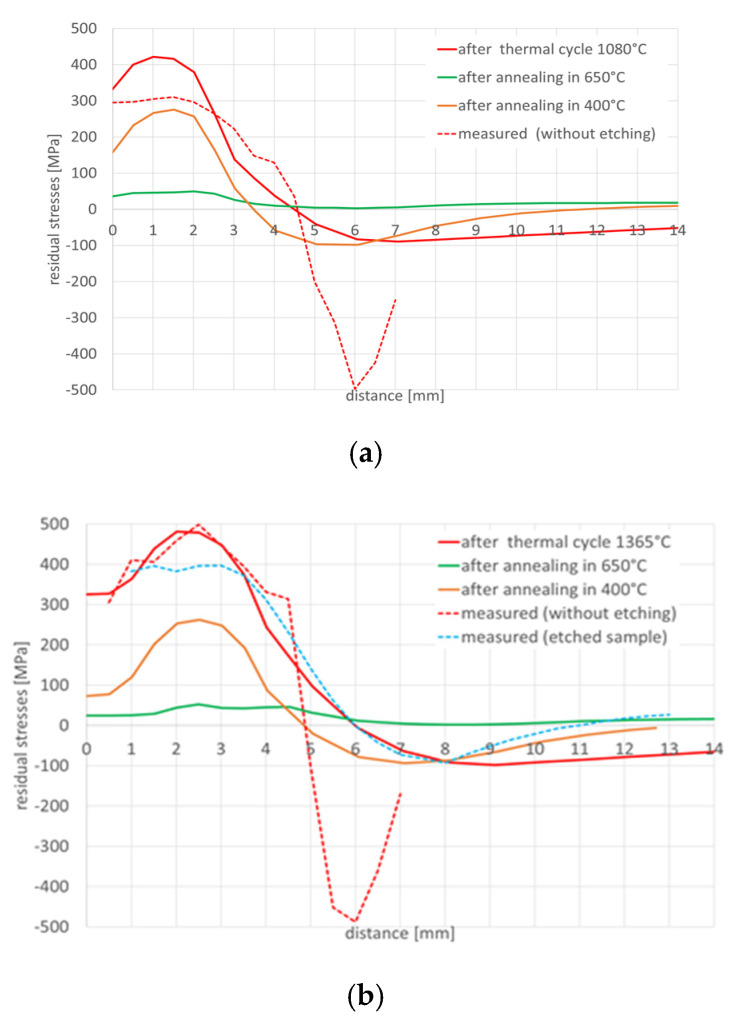
Mean stresses distribution graphs after annealing in 400 °C and 650 °C by 2 h for (**a**) 1360 °C and (**b**) 1080 °C maximal temperature of welding thermal cycles.

**Figure 18 materials-13-05289-f018:**
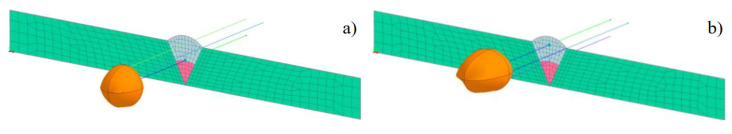
A view of 2D cross-section model of butt joint multipass GMAW welded and heat source models used for (**a**) 1st and (**b**) 2nd bead welding with visible double-ellipsoidal heat source models.

**Figure 19 materials-13-05289-f019:**
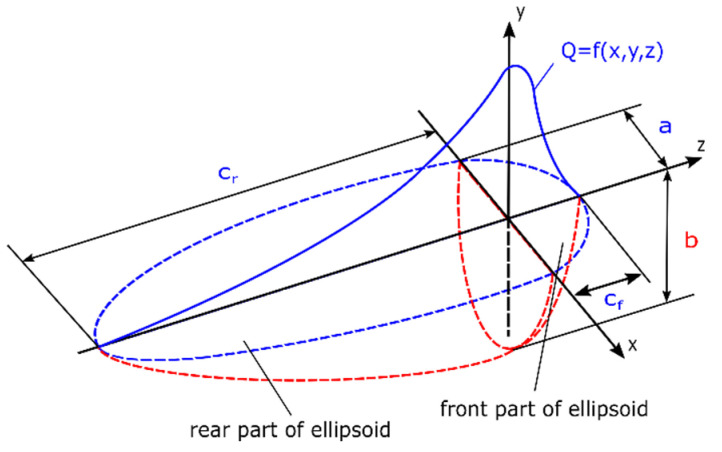
A view of the double-ellipsoidal (Goldak’s) heat source model [31,33].

**Figure 20 materials-13-05289-f020:**
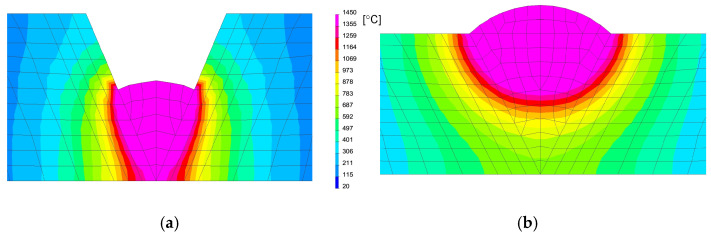
A view of temperatures distribution during (**a**) 1st and (**b**) 2nd bead welding.

**Figure 21 materials-13-05289-f021:**
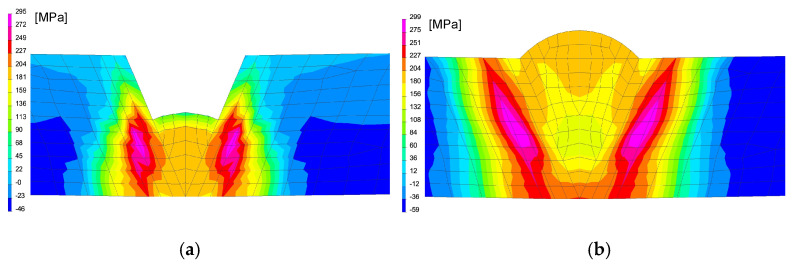
A view of the residual mean stresses distribution after (**a**) 1st and (**b**) 2nd bead welding and cooling to the ambient temperature.

**Figure 22 materials-13-05289-f022:**
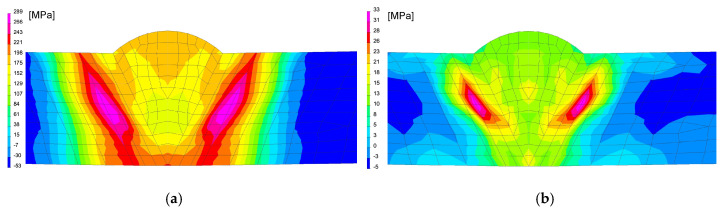
A view of the residual mean stresses distribution after annealing by 2 h at temperatures: (**a**) 400 and (**b**) 650 °C.

**Figure 23 materials-13-05289-f023:**
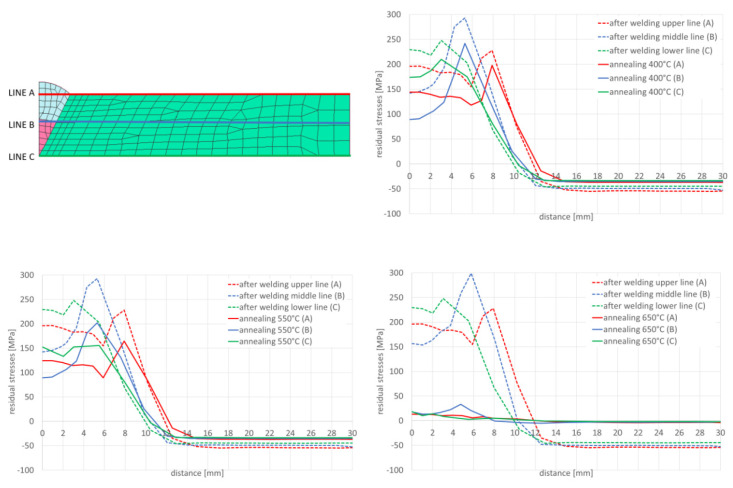
Graphs of mean stresses values in the multipass butt-joint GMA welded, depending on the distance from the weld axis for annealing temperatures equal to 400, 550 and 650 °C.

**Figure 24 materials-13-05289-f024:**
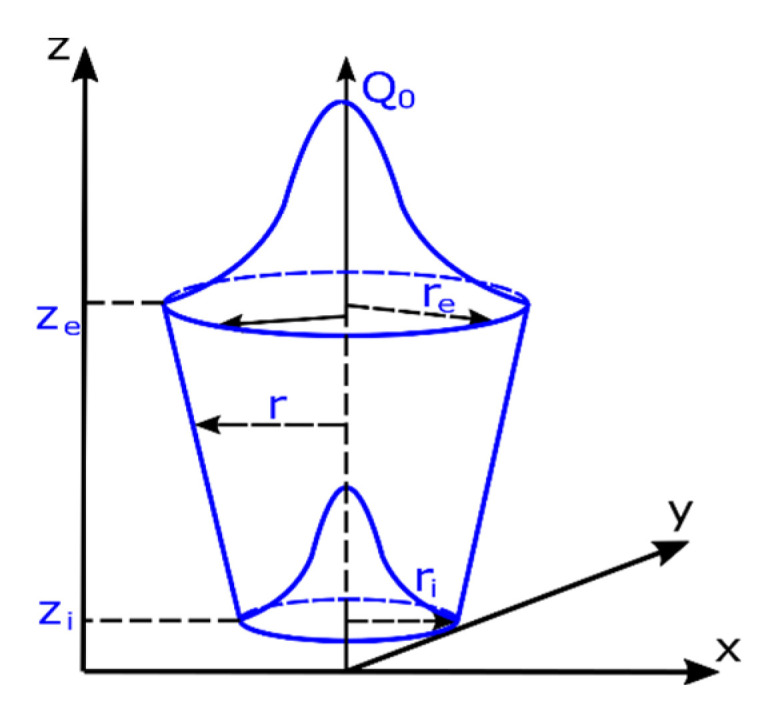
A view of the 3D conical heat source model [30,32].

**Figure 25 materials-13-05289-f025:**
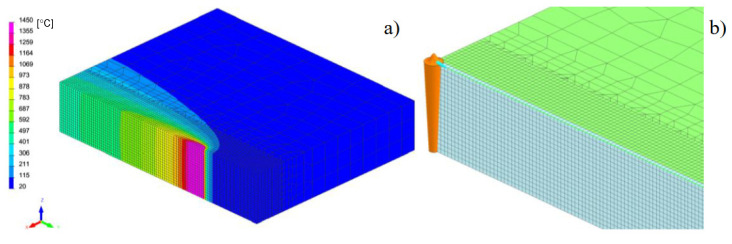
A view of (**a**) temperatures distribution during weld and (**b**) 3D conical heat source model.

**Figure 26 materials-13-05289-f026:**
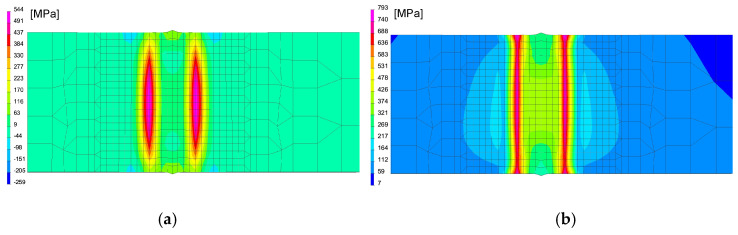
A view of distributions of (**a**) residual mean stresses and (**b**) VonMises stresses after welding on the specimen cross-section in the middle of model length.

**Figure 27 materials-13-05289-f027:**
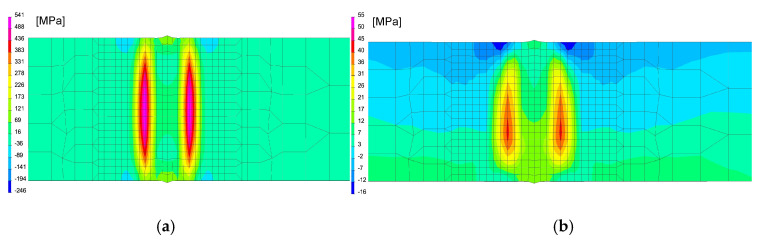
A view of the distribution of (**a**) residual mean stresses and (**b**) VonMises stresses after annealing 400 °C and 650 °C on the specimen cross-section in the middle of the model length.

**Figure 28 materials-13-05289-f028:**
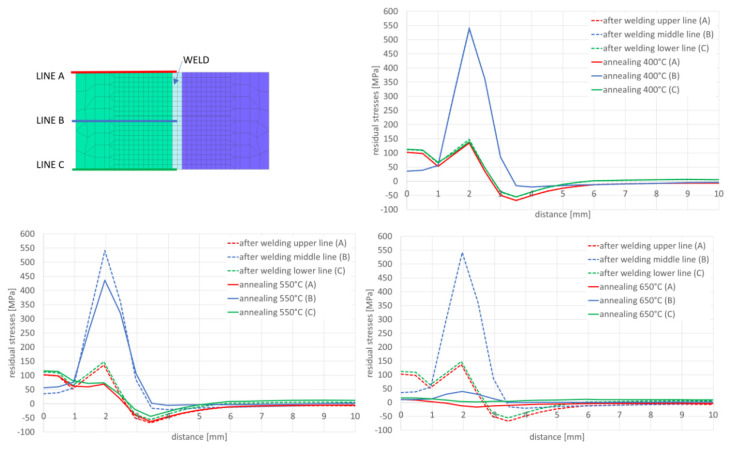
Graphs of residual mean stresses values in the laser welded butt-joint, depending on the distance from the weld axis for annealing temperature equal to: 400, 550 and 650 °C.

**Figure 29 materials-13-05289-f029:**
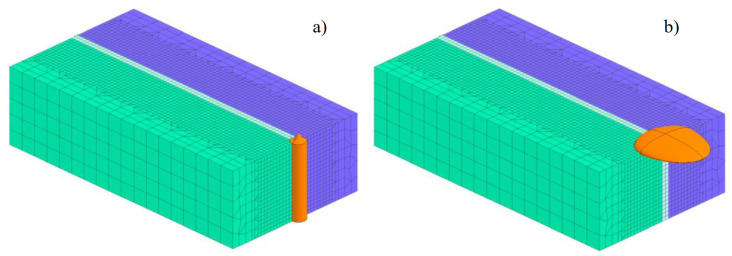
A view of 3D model of butt joint hybrid welding and heat source models used for (**a**) laser and (**b**) arc welding (MAG).

**Figure 30 materials-13-05289-f030:**
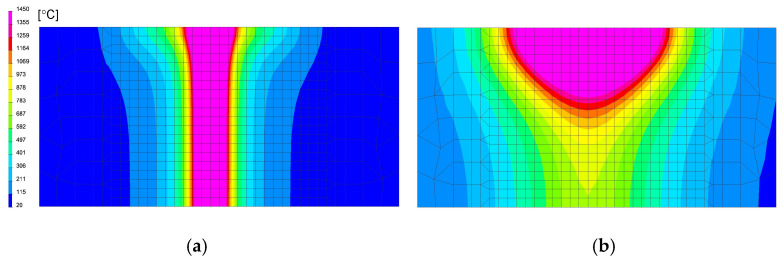
A view of temperature fields on the cross-section of hybrid welded butt joint for: (**a**) laser beam and (**b**) arc welding heat source.

**Figure 31 materials-13-05289-f031:**
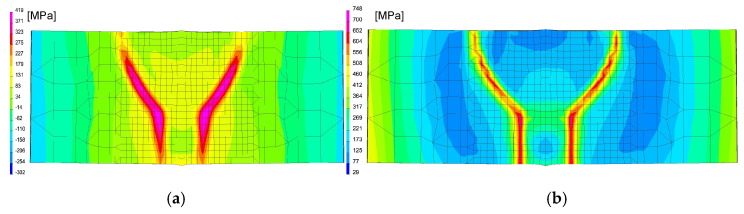
A view of distribution of (**a**) residual mean stresses and (**b**) VonMises stresses after welding on the specimen cross-section in the middle of model length.

**Figure 32 materials-13-05289-f032:**
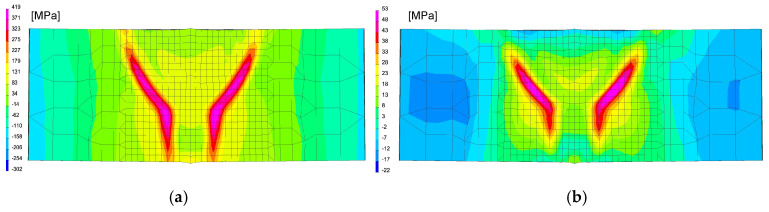
A view of residual mean stresses distributions after welding and annealing in: (**a**) 400 °C and (**b**) 650 °C by 2 h.

**Figure 33 materials-13-05289-f033:**
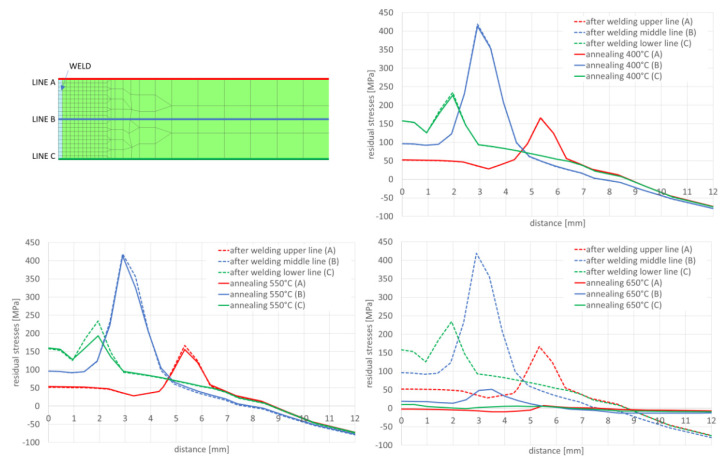
Graphs of residual mean stresses values in the hybrid welded butt-joint, depending on the distance from the weld axis for annealing temperature equal to 400 °C, 550 °C and 650 °C.

**Table 1 materials-13-05289-t001:** Chemical composition according to EN 10149-2 and measured on S700MC steel samples.

Chemical Composition, % wt.										
C max.	Si max.	Mn max.	P max.	S max.	Al min.	Nb max. *	V max. *	Ti max. *	B max.	Mo max.
0.12	0.60	2.10	0.008	0.015	0.015	0.09	0.20	0.22	0.005	0.50
Chemical composition, % wt. measured on spectrometer										
0.051	0.197	1.916	0.006	0.006	0.038	0.063	0.072	0.056	0	0.113

* Total content of Nb, V and Ti should not exceed max. 0.22%.

**Table 2 materials-13-05289-t002:** Results of mechanical properties tests of used S700MC steel.

Specimen Designation	Specimen Diameter [Mm]	Yield Strength R_e_ [MPa]	Tensile Strength R_m_ [MPa]	Elongation A_g_ [%]	Elongation A_40_ [%]
S1	6.52	759	851	10.97	23.91
S2	6.53	758	852	12.33	25.22
S3	6.50	741	849	10.72	23.62
Average value:		753 ± 8.0	851 ± 1.0	11.01 ± 0.71	24.25 ± 0.7

Remarks: A_g_—percentage non-proportional elongation at maximum force.

**Table 3 materials-13-05289-t003:** Maximum values of residual stresses after annealing by 2 and 4 h at selected temperatures.

Sample Designation	Annealing Temperature (2 h Heating)				
	BM/200	BM/300	BM/400	BM/500	BM/550
AN1_2	508/413	468/306	431/218	388/107	525/62
AN2_2	472/403	435/269	431/198	330/117	456/59
	**Annealing Temperature (4 h Heating)**				
	BM/200	BM/300	BM/400	BM/500	BM/550
AN1_4	499/416	545/323	341/184	538/108	500/9
AN2_4	382/307	540/315	341/196	489/108	482/4

Remarks: designation—BM/200—max. value of stress [MPa] in base material/after annealing at 200 °C etc.

**Table 4 materials-13-05289-t004:** Mechanical properties of the sample annealed at 450 °C for 2 and at 550 °C for 4 h.

Sample Designation	Sample Diameter D_o_ [mm]	Lower Yield Point R_el_ [MPa]	Upper Yield Point R_eh_ [MPa]	Tensile Strength R_m_ [MPa]	Elongation A_g_ [%]	Elongation A_40_ [%]
annealed at 450 °C for 2 h						
T1_2	6.49	724.39	743.50	807.42	9.36	21.21
T2_2	6.52	744.01	773.54	825.36	9.38	20.88
annealed at 550 °C for 4 h						
T1_4	6.49	722.66	745.38	784.61	9.51	20.93
T2_4	6.49	721.04	742.98	786.19	9.77	21.35

Remarks: A_g_—Percentage non-proportional elongation at maximum force.

**Table 5 materials-13-05289-t005:** Welding parameters used in numerical simulations of multipass butt joint GMAW welding.

Bead Number	EPUL (J/mm)	v (mm/s)	k	Heat Source Model Dimensions *
1st bead	800	5.0	0.8	10.0/10.0/5.0
2nd bead	1000	4.0	0.8	15.0/12.0/5.0
cooling conditions: free air in temperature 20 °C, preheating temperature: 65 °C				

Remarks: EPUL—energy per unit length, v—welding speed, k—energy efficiency factor; *—in sequence: molten pool length/molten pool width/penetration depth, as can be seen in Figure 19.

**Table 6 materials-13-05289-t006:** Welding parameters used in a laser butt-joint welding simulation, Figure 25.

Designation	EPUL (J/mm)	v (mm/s)	k	Heat Source Model Dimensions *
laser	360	16.0	0.8	2.0/1.5/10.0
cooling conditions: free air in temperature 25 °C, preheating temperature: 65 °C				

Remarks: EPUL—energy per unit length, v—welding speed, k—energy efficiency factor; *—for 3D conical model in sequence: top diameter/bottom diameter/penetration depth.

**Table 7 materials-13-05289-t007:** Welding parameters used in a hybrid butt-joint welding simulations, Figure 30.

Designation	EPUL (J/mm)	v (mm/s)	k	Heat Source Model Dimensions *
3D conical (laser)	360	16.0	0.8	2.0/2.0/10.0
double ellipsoidal (MAG)	490		0.6	9.0/8.0/2.0
cooling medium: free air in temperature 20 °C, preheating temperature: 65 °C				

Remarks: EPUL—energy per unit length, v—welding speed, k—energy efficiency factor; *—for laser model in sequence: top diameter/bottom diameter/penetration depth, as can be seen in Figure 24; for MAG model: molten pool length/width/penetration depth, as can be seen in Figure 19.

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
