# Peer review of "Experiments and Numerical Simulations of the Annealing Temperature Influence on the Residual Stresses Level in S700MC Steel Welded Elements"

_materials, 2020, doi:10.3390/ma13225289_

Round 1

Reviewer 1 Report

Unfortunately, in my opinion, this paper cannot be considered for the publication on the journal for the following reasons:

1) The novelty of the work is not clear;

2) The analysis of the state-of-the-art on the topics discussed by the authors is too poor;

3) The experimental tests and tools described in the paper are not new.

Author Response

Thank you for expressing your opinion on our work. However, it seems that the article may be very interesting for the reader. This work presents a comprehensive study on the effects of annealing temperature on the residual stresses level in S700MC steel welded joints. Both experiments and simulations were performed in this investigation. The experiments were designed carefully mimicking the welding thermal cycles, so the obtained test results are valuable for the research of residual stress in welds. The numerical simulations further present the distributions of residual stresses in samples similar to the test samples, as well as some welding joints in industry, showing the applicability of the present simulation method.

We do not fully agree with the statement that the research methods used and the proposed research are not new. For example, the approach to tests on a thermal cycle simulator (the shape of samples and the method of testing) or the combination of numerical analyses with the experiment is new, which allowed for the development of a methodology for conducting numerical analyses of annealing processes for elements welded from S700MC steel.

We strongly believe that the knowledge contained in the presented research and analyses will allow the reader to obtain new information on both the possibilities of modern software to carry out numerical analyses of welding and heat treatment processes as well as the preparation/support of own experiments, if necessary.

During a literature analysis at the stage of preparing the work, we found that it is difficult to find the results of similar works that would comprehensively combine the results of experiments with the results of numerical analyses of the annealing process of materials such as thermomechanically rolled steels. Therefore, we hope that we will manage to convince you to change the previous decision after all.

Reviewer 2 Report

Experiments and numerical simulations of the annealing temperature influence on the residual stresses level in S700MC steel welded elements is very interesting paper. Some minor changes are required:

Line 107:  VisualWeld (SYSWELD) Software package (Producer, Country)

Line 107: The SYSWELD solver (solver or Software?)

Line 114: In which temperature range is this equation 1 valid?

Line 160: Chemical composition, % wg (percentage by weight abbreviated wt %)

Line 243, 4·10-3 Torr (Can you  this value  in "Pa" convert/at Figure 4 "MPa" was used)

Conclusion:

Line 666, 667: The comparison of the obtained results of the analyses with the measurements carried out on the actual samples also showed another regularity. Why?

Author Response

Thank you for reviewing our article. In the attached pdf file we send answers to the questions contained in the review. We believe that the corrections introduced meet the requirements and the work is currently better.

Reviewer 3 Report

This manuscript presents a comprehensive study on the effect of annealing temperature on the residual stress level in S700MC steel welded elements. Both experiments and simulations were performed in this investigation. The experiments were designed carefully mimicking the welding thermal cycles, so the obtained test results are valuable for the research of residual stress in welds. The numerical simulations further present the distributions of residual stresses in samples similar to the test samples, as well as some welding joints in industry, showing the applicability of the present simulation method. However, it seems that the validity of the numerical simulation by experiments lacks and a careful comparison on the details of the results of residual stress between tests and simulations is needed. The following points are suggested to be considered.

  1. In Fig.3, the author aims to present the real welding thermal cycles and the simulated cycles. However, the thermal cycles in real welding cannot be found, which is important to clearly described. Furthermore, three questions related to this figure are not clear. 1) how did the thermal cycles in Fig.3 were measured or designed? 2) why the free cooling curve is lower than the controlled cooling? 3) How did "the controlled cooling" be controlled?
  2. Line 305, how did you know the depth of the high values of residual stresses is ~0.1 mm? If this was obtained from Fig.7, it is wrong because the distance of 0.1 mm at the maximum residual stress is the lateral distance from the center position, rather than the depth.
  3. Is it necessary to measure the initial state of residual stress of the sample after etching while before simulated welding thermal cycle? The reason for doing this test is to explain the great difference of the profile of residual stress between the specimens 3 and 5, especially at the position ~5 mm from the center, as shown in Fig.7.
  4. The Elongation was labeled with different symbols of A20 in Table 2 and A40 in Table 4, what are the differences between them?
  5. What are the annealing temperature for the T1_4 and T2_2 samples in Table 4? 450 or 550 degrees? This is questioned because in the legend it is 450 deg. while in the text it is 550 deg. Moreover, if it is 550 deg., why the yield stress for these two samples are similar to the samples annealed at 450 deg. for much less time?
  6. The comparison on residual stress profile between FEM simulation results and experimental results lacks. Furthermore, when comparing the two results such as Fig.9 and Fig. 17 for residual stress profile after thermal cycle 1080 degrees, one can find the FEM simulation does not show the peak compressive stress which was measured in experiments of ~500 MPa, as shown in Fig. 9.
  7. The conclusion part is too long and one cannot easily capture the main findings of this work.
  8. The unit for the stress in the simulation figures are missing, and the figures are not so clear.
  9. some English errors, such as: line 156 "values of yield strength and yield strength", line 386 "it is not it's so obvious"

Author Response

(The authors gave the same response as above.)

Round 2

Reviewer 1 Report

The authors did a good job on the revised manuscript.
Accept in its current form.

Author Response

Thank you for your opinion about our work, we tried to make every effort to ensure that the corrected version did not contain errors in the original.

Thank you for your time and please stay safe and healthy. 

Reviewer 3 Report

It is much clearer for the reviewer after reading the explanations and responses from the authors. However, it is still not so clear for readers because the authors revised not enough in the article. For example, the following points related to the comments #1, #2, #3 and #6 in the first reviewing report should be considered:

1. The legend of Fig.3 may mislead readers because the phrase of "comparison ... with..." may not be correctly used.

2. For the 0.1 mm depth in line 305, the reviewer suggests the authors to add a simple description of the method for obtaining this result.

3. Fig.7 compares the residual stresses distribution of two specimens, No. 3 and No. 5, and a big difference in the compressive stress peak can be found. Comparing the state of the specimens No.3 and No.5, it is found that: 1) specimen No. 3 was annealed, and then applied a thermal cycle; 2)specimen No. 5 was etched, and then applied a thermal cycle. If the etching has the same effect like annealing, there should be no such a big difference. Then to study the reason, the reviewer suggested to perform a XRD scan for the sample only after etching but without thermal cycle to see the effect of etching on the residual stress distribution in the first review report. However, in the revised manuscript, the reason for the big difference between specimens No. 3 and No. 5 remains unexplained.

4. The Response 6 is suggested to be included in the article.

Author Response

Once again, we would like to thank you for the enormous amount of work put into the review of our article and a number of very valuable tips and corrections. We hope that the following changes will make the article meet the expectations of the reviews.

Thank you for your time and please stay safe and healthy.

Please find also below responses to comments, which we already included in the corrected text of the article.

It is much clearer for the reviewer after reading the explanations and responses from the authors. However, it is still not so clear for readers because the authors revised not enough in the article. For example, the following points related to the comments #1, #2, #3 and #6 in the first reviewing report should be considered:

  1. The legend of Fig.3 may mislead readers because the phrase of "comparison ... with..." may not be correctly used.

Response 1:

The legend of the Fig. 3 was changed as follows:

Figure 3. Comparison of controlled cooling thermal cycles (green curve) which simulate welding cycle with thickness of 10 mm and thermal cycle with higher cooling rate (red curve) which simulate welding cycle with thickness higher than 10 mm

  1. For the 0.1 mm depth in line 305, the reviewer suggests the authors to add a simple description of the method for obtaining this result.

Response 2:

A description as follows: “In order to remove the thin surface layer, closed circuit electrolytic etching with the Electrolyte type A supplied by Proto Manufacturing Europe was used. Due to the constant flow of electrolyte, the etched area was not thermally affected” – was already added into text

  1. Fig.7 compares the residual stresses distribution of two specimens, No. 3 and No. 5, and a big difference in the compressive stress peak can be found. Comparing the state of the specimens No.3 and No.5, it is found that: 1) specimen No. 3 was annealed, and then applied a thermal cycle; 2)specimen No. 5 was etched, and then applied a thermal cycle. If the etching has the same effect like annealing, there should be no such a big difference. Then to study the reason, the reviewer suggested to perform a XRD scan for the sample only after etching but without thermal cycle to see the effect of etching on the residual stress distribution in the first review report. However, in the revised manuscript, the reason for the big difference between specimens No. 3 and No. 5 remains unexplained.

Response 3:

Etching does not have the same effect as annealing. During annealing, residual stresses in the entire cross-section of the sample are minimized, not only in the surface layer affected by machining. Only the affected surface layer is removed by etching, but the residual stresses after thermomechanical processing remain in the entire cross section of the sample.

We have also added this sentence in Conclusions that explains the difference – line 630.

  1. The Response 6 is suggested to be included in the article.

Response 4:

As suggested, it was added in the paragraph before Figure 15 as a continuation of the description of the issue mentioned – line 445.